# Layered Extraction and Adsorption Performance of Extracellular Polymeric Substances from Activated Sludge in the Enhanced Biological Phosphorus Removal Process

**DOI:** 10.3390/molecules24183358

**Published:** 2019-09-16

**Authors:** Daxue Li, Hailing Xi

**Affiliations:** 1State Key Laboratory of NBC Protection for Civilian, Beijing 102205, China; lidaxue678@163.com; 2Department of Military Installations, Army Logistic University of PLA, Chongqing 401331, China

**Keywords:** enhanced biological phosphorus removal (EBPR), tightly bound extracellular polymeric substances (TB-EPS), loosely bound extracellular polymeric substances (LB-EPS), phosphate, adsorption performance

## Abstract

A large amount of phosphorus was found in the extracellular polymeric substances (EPS) of activated sludge used in enhanced biological phosphorus removal (EBPR), so the role of EPS and extracellular phosphorus in EBPR should not be neglected. The composition and properties of tightly bound EPS (TB-EPS) and loosely bound EPS (LB-EPS) were significantly different, and it was necessary to study the adsorption performance of EPS through the fractionating of activated sludge into LB-EPS, TB-EPS and microbial cells. In this study, the adsorption performance of LB-EPS and TB-EPS for phosphate was explored by extracting LB-EPS and TB-EPS via sonication and cation exchange resin (CER), respectively. The results indicated that the sonication-CER method was an efficient and reliable extraction method for EPS with a synergistic effect. The performance of EPS in the adsorption/complexing of phosphate was excellent because of its abundant functional groups. Specifically, the type and content of metal elements and functional groups in TB-EPS were much greater than those in LB-EPS, which led to the key role of TB-EPS in the adsorption/complexing of phosphate. Finally, a metabolic model for EBPR with consideration of the adsorption performance of LB-EPS and TB-EPS was proposed.

## 1. Introduction

Extracellular polymeric substances (EPS) are widely distributed in biological aggregates, which has an important influence on the wastewater biological treatment process [1,2]. In the past two decades, EPS have been shown to contain a large amount of phosphorus, which is the phosphorus storage and transfer station in the enhanced biological phosphorus removal (EBPR) process [3]. Importantly, the role of EPS in the EBPR process is nonnegligible and should be considered in the mechanisms of phosphorus removal. In activated sludge, microbial cells are embedded within EPS, which can be divided into tightly bound EPS (TB-EPS) located in the inner layer of the sludge floc and loosely bound EPS (LB-EPS) existing in the outer layer of the sludge floc [1,4]. The contents, compositions and properties of TB-EPS and LB-EPS were found to be different [5,6,7], and their influences on the flocculation, sedimentation and dewatering of activated sludge were also significantly different. To acquire more information about the exact role of EPS in EBPR, it is necessary to fractionate EPS into TB-EPS and LB-EPS.

There are many methods to extract EPS from EBPR sludge, including sonication, cation exchange resin (CER), sonication-CER, heating, alkali treatment, ethylenediaminetetraacetic acid (EDTA), and formaldehyde/NaOH treatment [8,9]. In recent years, the sonication method has been gradually favored by researchers. However, the biggest limitation of sonication for extracting EPS is a low extraction efficiency, but the sonication-CER method can solve this problem [4,10,11]. The content and species of extracellular phosphorus are seldom affected by intracellular phosphorus in the extraction process with CER or sonication [9], which is the basis of accurate analysis of extracellular phosphorus. The phosphorus content of EPS can reflect the role of EPS in biological phosphorus removal. Zhang et al. [3] compared the formaldehyde-NaOH extraction method with sonication, EDTA, heating and CER extraction, and the results showed that CER extraction was more suitable for extracting EPS from EBPR sludge. Moreover, although the formaldehyde-NaOH, EDTA and heating extractions had higher EPS extraction efficiencies, these methods caused chemical pollution and bacterial cell lysis. Therefore, the extraction of extracellular polymers from EBPR sludge by sonication and CER is an effective way to study the adsorption ability.

EPS are a kind of macromolecular mixture produced by microorganisms under specific conditions, the components of which are mainly proteins, humus and polysaccharides [8]. Due to the existence of these macromolecular substances, EPS have a large surface area and many functional groups, including carboxyl, phosphate, sulfate, amino, phenol, hydroxyl and other active groups [12]. Many functional groups in EPS can adsorb heavy metals by electrostatic interactions or complexation. In addition, hydrophobic regions exist in EPS, which can adsorb nonpolar organic pollutants. With the above characteristics, the adsorption capacity for heavy metals and organic pollutants in wastewater is strong [13,14]. The interaction between phosphorus and EPS is beneficial for the accumulation of polyphosphate (polyP) in EPS. The content of extracellular phosphorus can reach nearly 10% of total phosphorus in EBPR-activated sludge [2]. Biological phosphorus removal from wastewater is mainly achieved by discharging phosphorus-rich activated sludge, so the combination of EPS and phosphorus will not only affect phosphorus absorption and utilization but also affect phosphorus removal in EBPR bioreactors [15].

In biological treatment systems, the type of organic substrate and temperature that have substantial effects on the microbial community and metabolism are important factors in biological phosphorus removal [16,17,18]. Acetate and propionate are the typical volatile fatty acids (VFAs) in sewage or municipal wastewater, which are key carbon sources for the EBPR process [19,20]. Temperatures higher than 20 °C were generally reported as unfavorable to the EBPR process, so the EBPR technique might not be suitable for tropical climates with temperatures above 30 °C [21,22,23,24]. Furthermore, the content and composition of LB-EPS and TB-EPS as well as their roles in biological phosphorus removal might also be influenced by the type of organic substrate and the temperature. However, information about the role of LB-EPS and TB-EPS in phosphate adsorption/complexing under different temperatures and substrate types has not been presented in the literature. This study aimed to investigate the roles of LB-EPS and TB-EPS in EBPR. For this purpose, four lab-scale Anacrobic/ Oxic Sequencing Batch Reactor (A/O-SBR) reactors with different temperatures (20 ± 1 °C or 35 ± 1 °C) and organic substrates (acetate or propionate) were operated under steady-state conditions. LB-EPS and TB-EPS were extracted by the sonication method and the CER method, respectively, and the contents and functional groups in sludge flocs were analyzed. The differences in adsorption performance between LB-EPS and TB-EPS were evaluated. Furthermore, the biochemical reaction process of TB-EPS phosphate adsorption/complexing in activated sludge was clarified to understand the roles of LB-EPS and TB-EPS in EBPR.

## 2. Results and Discussion

### 2.1. Extraction Efficacy of EPS

The extraction efficacy of EPS via sonication-CER was better than that with sonication or CER alone. From Figure 1, the extraction amounts of EPS with sonication, CER, and sonication-CER were 11.75~28.06 mg total organic carbon (TOC)/g volatile suspended solid (VSS), 36.64~138.15 mg TOC/g VSS, and 51.26~168.27 mg TOC/g VSS, respectively, which were equivalent to 2.9%~6.4%, 8.9%~31.5%, and 12.4%~38.4% of TOC_sludge_, respectively. For the four kinds of activated sludge, the amount of EPS extracted via sonication-CER was greater than the sum of that extracted with sonication and CER individually, which suggested that there was a synergistic effect when extracting EPS via sonication-CER.

As shown in Figure 2, the amount of TB-EPS extracted from activated sludge was much higher than that of LB-EPS, mainly because TB-EPS was a dense layer close to the outside of the microbial cell membrane and contained protein molecules with higher molecular weight. The content of LB-EPS and TB-EPS in activated sludge with sodium acetate or sodium propionate as the sole carbon source at 20 °C was greater than that at 35 °C, indicating that the net production of LB-EPS and TB-EPS in activated sludge at high temperature (35 °C) was lower. Furthermore, the changes in the TB-EPS content of activated sludge at 20 °C were less than those at 35 °C during the anaerobic–aerobic cycle, indicating that the inner structure of the activated sludge cultured at 20 °C was more stable.

### 2.2. Phosphorus Removal Performance and Microbial Ecological Mechanism

The biological phosphorus removal performances of the 4 A/O-SBR reactors were distinctly different (Figure 3). From Figure 3, the profiles of total phosphorus (TP) in the supernatant of the 20 °C reactors presented a typical EBPR pattern, with a considerable transportation of TP from the sludge to the bulk solution in the anaerobic period and an opposite flow in the aerobic period. Nevertheless, the activated sludge in the 35 °C reactors had non-EBPR performance, with a relatively stable value of TP in solution during the anaerobic–aerobic cycle. The above results show that temperature has an important influence on the biological phosphorus removal performance of activated sludge, which is consistent with the work published by Kee et al. [21] and Sayi et al. [22]. Phosphorus-accumulating organisms (PAOs) are the key functional microorganisms in EBPR and play an important role in biological phosphorus removal [25]. According to the microscope images acquired with scanning electron microscopy (SEM) (see Figure 4), the main microorganisms in the sludge of the two 20 °C reactors were rod-shaped bacteria, which are common PAOs and play a major role in EBPR. Nevertheless, the main microorganisms in the sludge of the two 35 °C reactors were filamentous bacteria, which had poor phosphorus removal performance, and sludge bulking occurred under deteriorating environmental conditions, which caused the irreversible collapse of the reactor.

### 2.3. Properties of LB-EPS and TB-EPS Phosphorous Adsorption/Complexing

As shown in Figure 5, the Fourier-transform infrared spectroscopy (FTIR) spectra of EPS in activated sludge of the two 35 °C reactors and the two 20 °C reactors were extremely different, indicating that the biochemical reaction temperature affected the type and content of functional groups of LB-EPS and TB-EPS. The difference in infrared spectra between LB-EPS and TB-EPS in the 35 °C reactors was mainly concentrated in the fingerprint region (<1000 cm^−1^), which indicated that there were significant differences in the amount of phosphorous-containing and sulfur-containing groups between LB-EPS and TB-EPS in activated sludge. The position and amount of absorption peaks in the functional group region (1000–2500 cm^−1^) of LB-EPS and TB-EPS were similar, and the absorption peaks at 3500–3300 cm^−1^, 1680–1630 cm^−1^, 1630–1580 cm^−1^, 1460–1350 cm^−1^, and 1000–1150 cm^−1^ were all apparent, which indicated that LB-EPS and TB-EPS in the activated sludge of the four reactors contained hydroxyl or amino groups, amide groups (protein peptide chain), polysaccharides or primary alcohols. The absorption peaks at 1330–1200 cm^−1^ of LB-EPS in the 35 °C reactors were extremely weak, while those of TB-EPS were obvious, demonstrating that the carboxylic acid, phenol and polysaccharide groups in TB-EPS were more abundant than those in LB-EPS. Furthermore, the absorption peaks at 1680–1630 cm^−1^ of TB-EPS in the 35 °C reactors were stronger than in TB-EPS, which indicated that the amide groups in TB-EPS were more plentiful than those in LB-EPS.

As shown in Figure 6, TP_EPS_/TOC_EPS_ of LB-EPS changed irregularly during the anaerobic–aerobic cycle, and TP_EPS_/TOC_EPS_ of LB-EPS in the activated sludge of the 20 °C sodium acetate reactor was significantly higher than that of the other three reactors. Moreover, in the same activated sludge, the phosphorous adsorption/complexing performance of TB-EPS was significantly higher than that of LB-EPS. Specifically, the TP_EPS_/TOC_EPS_ of TB-EPS in activated sludge from the 35 °C sodium acetate reactor and sodium propionate reactor and from the 20 °C sodium acetate reactor and sodium propionate reactor were 2.79–4.97, 1.56–2.63, 1.07–2.88 and 1.70–2.98 times higher than those of LB-EPS, respectively. The above phenomenon is mainly explained by two aspects. LB-EPS and TB-EPS were extracted via sonication and CER, respectively, and CER was more efficient at extracting phosphorus from EPS than sonication. Furthermore, the affinity for phosphate of TB-EPS was stronger than that of LB-EPS.

### 2.4. The Effect of Metal Elements on TB-EPS Phosphorous Adsorption/Complexing

As shown in Figure 7, the content of Ca in TB-EPS from activated sludge was much greater than that of other metals in the 35 °C reactors, and the content of K, Mg and Ca in TB-EPS from activated sludge in the 20 °C reactors was greater than that of Al and Fe. Fe in activated sludge mainly existed in microbial cells, while most Al was located in TB-EPS [26]. The main species of phosphorus (P) in the activated sludge of the 35 °C reactors was orthophosphate (orthoP) [27] and a small number of orthoP combined with K/Mg to form active metabolic complexes, while most orthoP combined with Ca/Al/Fe to form inert metabolic complexes. In addition, the Ca/Al-P complex mainly existed in TB-EPS, and the Ca/Fe-P complex was mainly located in microbial cells, both of which did not participate in the microbial metabolism process. The contents of K and Mg in TB-EPS from activated sludge were both below 0.5 mg/g TOC, the variation trends of which were not anaerobic-decreasing and aerobic-increasing during the anaerobic–aerobic cycle. TB-EPS in the activated sludge of the 35 °C reactors directly released or took up orthoP, which played the role of the phosphorus (P) transfer station. Therefore, a certain amount of phosphorus was maintained in the activated sludge of the 35 °C reactors without biological phosphorus accumulation, and the EBPR performance was poor.

The main species of P in activated sludge from the 20 °C reactors were orthoP and polyphosphate (polyP), and polyP was mainly located in TB-EPS [27]. Some P (including polyP and orthoP) in the activated sludge combined with K and Mg to form metabolic active complexes, such as K/Mg-polyP and K/Mg-orthoP, which could promote the migration and transformation of P in activated sludge. In addition, K/Mg-polyP mainly existed in TB-EPS, and K/Mg-orthoP was mainly located in microbial cells. Furthermore, the other portion of P (mainly orthoP) combined with Ca/Al/Fe to form inert metabolic complexes, in which Ca/Al-P complexes mainly existed in TB-EPS and Ca/Fe-P complexes were mainly located in microbial cells. Due to the high content of K and Mg in the activated sludge of the 20 °C reactors, a large amount of P combined with K and Mg, and the main phosphorus species of TB-EPS were polyP. Moreover, the variation trends of K and Mg in TB-EPS from the activated sludge in the 20 °C reactors were anaerobic-decreasing and aerobic-increasing, which indicated that TB-EPS participated in the biological phosphorus accumulation process. Therefore, the EBPR performance of activated sludge of the 20 °C reactors was good.

### 2.5. The Biochemical Reaction Process of TB-EPS Phosphate Adsorption/Complexing in EBPR-Activated Sludge

As shown in Figure 8, the biochemical reaction process of TB-EPS phosphate adsorption/complexing in EBPR-activated sludge involved the migration and transformation between metal elements, polyP, proteins, and polysaccharides, which was closely related to anaerobic/aerobic metabolism. At the anaerobic stage (Figure 8a), the microbial cell decomposed intracellular energy substances (mainly intracellular glycogen) and released metabolic secretions into TB-EPS and LB-EPS. In addition, these metabolic secretions were mainly proteins and polysaccharides, in which the protein content was larger than that of polysaccharides. The microbial cells released intracellular K/Mg-orthoP into EPS and the bulk solution. Meanwhile, long-chain K/Mg-polyP in TB-EPS was decomposed into short-chain polyP and orthoP. The abovementioned orthoP accomplished anion exchange with VFAs in the bulk solution, which promoted the transfer of VFAs from the bulk solution into microbial cells to be synthesized into polyhydroxyalkanoates (PHA) with the energy produced by intracellular glycogen. In addition, the orthoP released by the microbial cell and produced by K/Mg-polyP decomposition in TB-EPS were transferred into the bulk solution, with the release of K^+^ and Mg^2+^. The above results indicated that the microbial cell (mainly phosphorus-accumulating organisms, PAOs) obtained the competitive advantage of a carbon source.

At the aerobic stage (Figure 8b), taking O_2_ as the electron acceptor, PHA in the microbial cell decomposed, and intracellular glycogen was synthesized. Meanwhile, orthoP and metal ions in the bulk solution were absorbed and transferred into the microbial cell, using some of the proteins and polysaccharides in LB-EPS and TB-EPS as carbon sources. OrthoP and metal ions were transferred into TB-EPS after a short residence time in LB-EPS. Then, orthoP combined with metal ions, transforming into the inert Ca/Al-orthoP complex and the active K/Mg-orthoP complex in TB-EPS. Some of the K/Mg-orthoP was stored in TB-EPS, and the remainder was transferred into the microbial cell. Moreover, orthoP, which was transferred from the bulk solution and EPS into the microbial cell and combined with intracellular K, Mg, Ca and Al, was transformed into the active K/Mg-orthoP complex and the inert Ca/Fe-orthoP complex. With the energy generated by PHA decomposition and with the consumption of intercepted K/Mg-orthoP, long-chain K/Mg-polyP was synthesized in TB-EPS.

## 3. Materials and Methods

### 3.1. Culture of Activated Sludge

The activated sludge of four lab-scale A/O-SBR reactors was fed synthetic wastewater, adopting sodium acetate or sodium propionate as the sole carbon source. The chemical oxygen demand (COD): nitrogen (N): phosphorus (P) ratio of the synthetic wastewater was 100:5:5, and the pH was approximately 7.0, and the trace elements are shown in Appendix A. The reactors had a working volume of 15 L and were operated for 2 cycles every day, adopting instantaneous wastewater filling. Each cycle lasted for 12 h, involving a 4-h anaerobic period, 7 h of aeration, a 50-min settlement time, 5 min of decanting, and 5 min of idling. The temperatures of the 4 reactors were controlled at 20 ± 1 °C or 35 ± 1 °C. The solid retention times (SRTs) were approximately 20 d, with discharging of the mixed liquid occurring every day. The dissolved oxygen (DO) concentrations under anaerobic conditions were 0.2–0.5 mg/L, which would be directly related to the growth status, phosphorus release capacity, and the ability to synthesize an organic matrix of phosphorus-accumulating organisms (PAOs). Phosphorus uptake by microorganisms under aerobic conditions was much higher than that released under anaerobic conditions. Fine air bubbles for aeration were supplied through a dispenser at the bottom of the reactor with an airflow rate of 1.6 L/h, which ensured that the DO concentration at the end of the aeration stage was 3.0–5.0 mg/L. The mixed liquid suspended solids (MLSS), sludge volume index (SVI) and the COD and TP of the effluent were monitored every day. After the values of the parameters remained relatively stable for 2 months, the experimental study was carried out.

### 3.2. Method of Washing CER

The 001 × 7 gel-type CER (16–40 mesh, Suqing, Jiangsu, China) was used for the CER extractions. The CER was washed successively with 8% NaCl, 1 mol/L HCl, and 1 mol/L NaOH, alternating the HCl and NaOH solutions for 3 washes, washing to a neutral pH by pure water each time. At the end of the washing process, two successive alkaline washes were implemented, after which the CER was washed to a neutral pH for use.

### 3.3. Extraction of EPS

#### 3.3.1. Extraction of LB-EPS

A modified sonication method was used to extract LB-EPS [4]. A sonicator operating at 21 kHz (JY90-II; Scientz Bioscience Co., Inc., Ningbo, China) was used to extract 40 mL sludge (VSS was between 7500 mg/L and 8500 mg/L) after centrifugation and resuspension. Subsequently, the treated sludge was centrifuged two times at 0 ± 2 °C and 43,000 RCF, and the twice centrifuged supernatant was designated LB-EPS. The sonication probe area was 0.28 cm^2^, the ultrasonic power density was 1 W/mL, and the process time was 6 min. The duty cycle during the sonication process was 50%.

#### 3.3.2. Extraction of TB-EPS

After extracting LB-EPS, the centrifuged pellet was resuspended in 40 mL water. Then, a modified CER method was used to extract TB-EPS [24]. The 001 × 7 gel-type CER (20–40 mesh, Suqing, Jiangsu, China) was employed to process the 40 mL mixed liquid samples. The amount of resin used was 100 g of CER/g VSS, and the reaction time was 30 min. Then, the CER was filtered using nylon mesh with a 250-mm pore diameter. Finally, the filtered mixed liquid was centrifuged two times at 0 ± 2 °C and 43,000 RCF, and the twice centrifuged supernatant was designated as TB-EPS.

#### 3.3.3. Extraction of EPS

The washed sludge was processed by sonication and then reacted with the CER to perform cation exchange. Thereafter, the filtered mixed liquid was centrifuged twice at 0 ± 2 °C and 43,000 RCF, and the supernatant was designated EPS.

### 3.4. FTIR Analysis

The functional groups constituting LB-EPS and TB-EPS extracted from sludge samples were well recognized by FTIR, which provides molecular information of bond angles and bonding patterns. The dried EPS was compressed to a 3 mm (diameter) disc using potassium bromide (KBr) and was recorded with a Bruker TENSOR 37 FTIR spectrophotometer at room temperature under inert conditions with a frequency range of 400 to 4000 cm^−1^ and at a resolution of 4 cm^−1^.

### 3.5. Other Analyses

The total organic carbon (TOC) of LB-EPS and TB-EPS extracts was detected using a TOC analyzer (multi N/C 2100 S, Analytik Jena AG, Jena, Germany). The DO concentration in mixed liquid was examined with a DO analyzer (Pro20, YSI, Columbus, Ohio, USA). The COD in effluent was detected using a COD instrument (DR1010, HACH, Loveland, Colorado, USA). The SVI, suspended solids (SS), volatile suspended solids (VSS), and TP in the bulk solution (TP_solution_), LB-EPS (TP_LB-EPS_), and TB-EPS (TP_TB-EPS_) were measured according to standard methods [28]. The TP in sludge (TP_Sludge_) was measured after resuspended sludge was dispersed by sonication, and the TP in microbial cells (TP_cell_) was calculated by subtraction. The metal elements in the bulk solution, sludge, LB-EPS, and microbial cells were directly measured by ICP-OES (Agilent 715, Agilent Technologies Inc., Palo Alto, CA, USA). The metal elements in TB-EPS were calculated by subtraction. The morphology of microorganisms in the activated sludge was observed by a scanning electron microscope (MIRA 3 GMU/GMH; TESCAN, Brno, Czechoslovakia).

## 4. Conclusions

In this work, the adsorption performance of LB-EPS and TB-EPS for phosphate was explored, in which sonication and CER were used to extract LB-EPS and TB-EPS, respectively, from the activated sludge of four lab-scale A/O-SBR reactors with different temperatures and organic substrates. The results revealed that the content and functional groups of LB-EPS and TB-EPS in activated sludge are distinctly different. More concretely, the content of hydroxyl, amino, amide (protein peptide chain), polysaccharide, and primary alcohol groups in TB-EPS was much higher than that in LB-EPS. Moreover, the phosphate adsorption performance of TB-EPS was better than that of LB-EPS. More importantly, the supposed metabolic model for EBPR with consideration of the adsorption performance of LB-EPS and TB-EPS was proposed, and the migration and transformation of metal elements, phosphate, proteins, and polysaccharides during the anaerobic–aerobic cycle were analyzed.

## Figures and Tables

**Figure 1 molecules-24-03358-f001:**
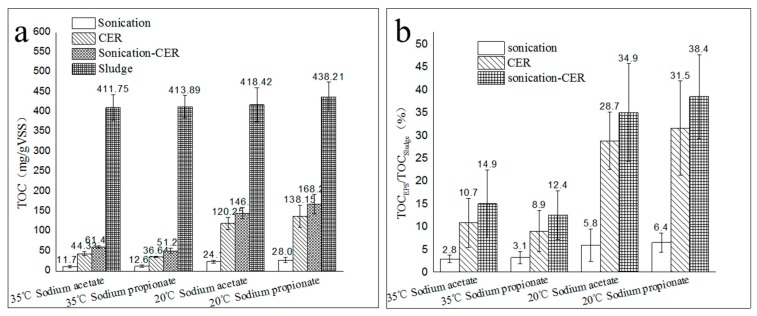
The extracted amounts of extracellular polymeric substances (EPS) when utilizing sonication, cation exchange resin (CER), and sonication-CER. (**a**): TOC_EPS_ and TOC_sludge_; (**b**): TOC_EPS_/TOC_sludge_ ratio. Abbreviations: TOC: total organic carbon; VSS: volatile suspended solid.

**Figure 2 molecules-24-03358-f002:**
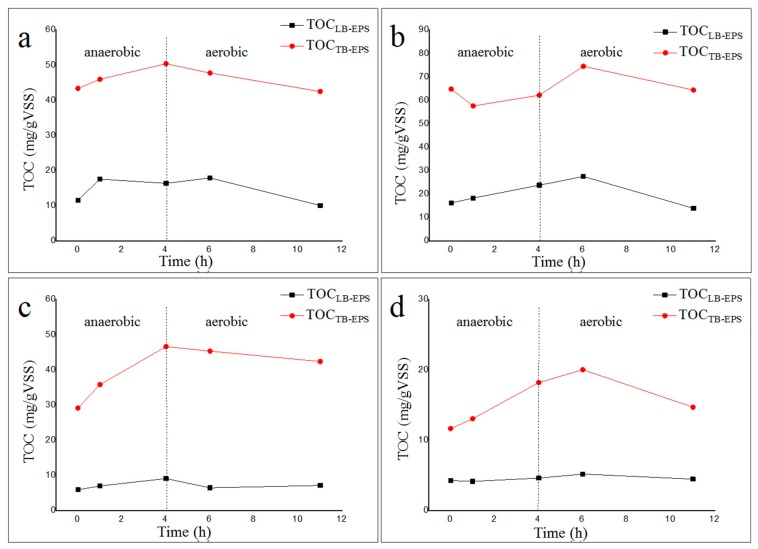
Content of loosely bound extracellular polymeric substances (LB-EPS) and tightly bound extracellular polymeric substances (TB-EPS) during the anaerobic–aerobic cycle. (**a**): LB-EPS and TB-EPS from a 20 °C sodium acetate reactor; (**b**): LB-EPS and TB-EPS from a 20 °C sodium propionate reactor; (**c**): LB-EPS and TB-EPS from a 35 °C sodium acetate reactor; (**d**): LB-EPS and TB-EPS from a 35 °C sodium propionate reactor.

**Figure 3 molecules-24-03358-f003:**
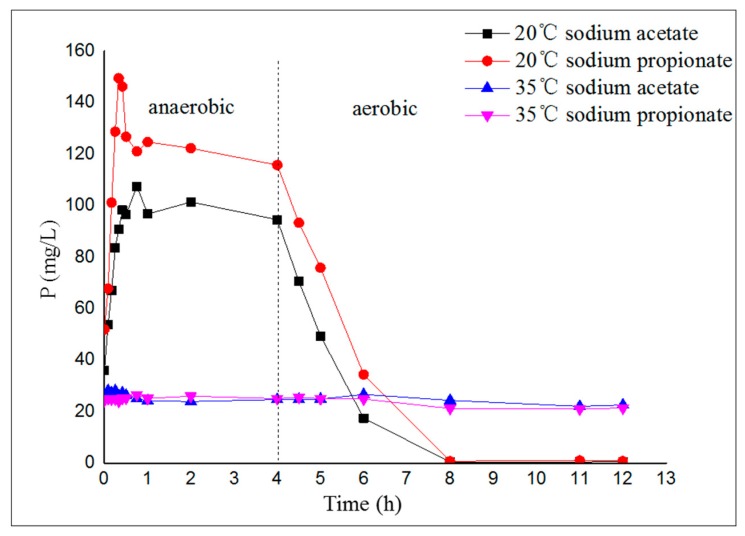
The variation of total phosphorus (TP) concentration in the supernatant during the anaerobic–aerobic cycle.

**Figure 4 molecules-24-03358-f004:**
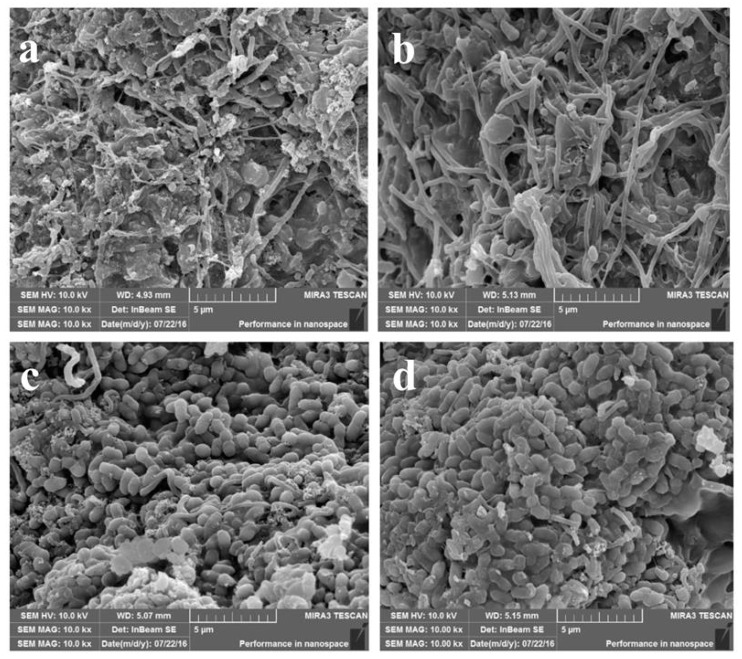
Microbial morphology of the four reactors observed by SEM. (**a**): Microbial morphology of a 35 °C sodium acetate reactor; (**b**): microbial morphology of a 35 °C sodium propionate reactor; (**c**): microbial morphology of a 20 °C sodium acetate reactor; (**d**): microbial morphology of a 20 °C sodium propionate reactor.

**Figure 5 molecules-24-03358-f005:**
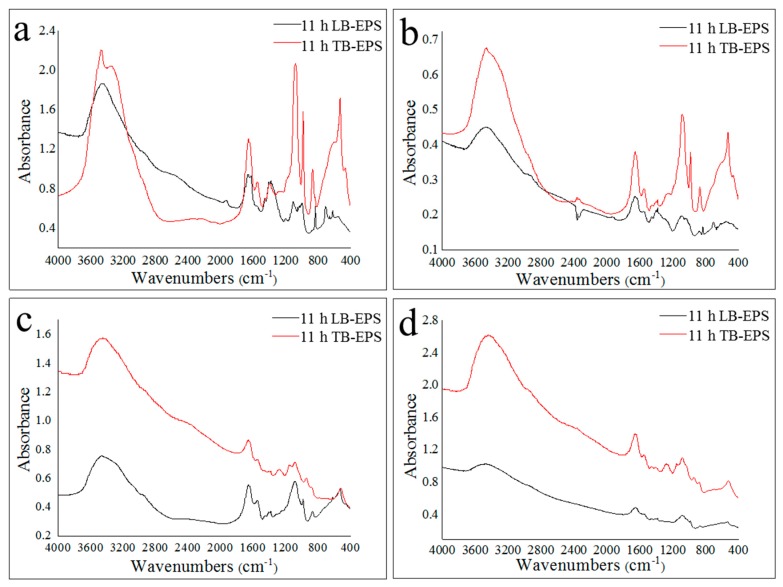
IR spectra of LB-EPS and TB-EPS at the later aerobic stage. (**a**): LB-EPS and TB-EPS from a 35 °C sodium acetate reactor; (**b**): LB-EPS and TB-EPS from a 35 °C sodium propionate reactor; (**c**): LB-EPS and TB-EPS from a 20 °C sodium acetate reactor; (**d**): LB-EPS and TB-EPS from a 20 °C sodium propionate reactor.

**Figure 6 molecules-24-03358-f006:**
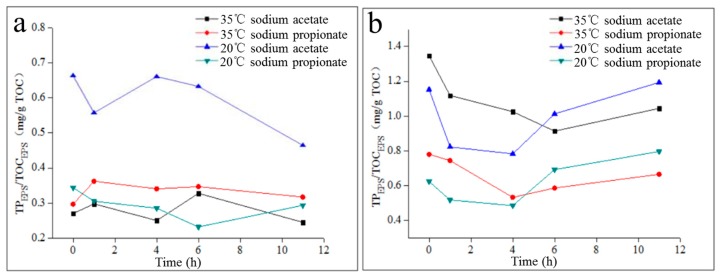
Performance of LB-EPS and TB-EPS phosphorous adsorption/complexing during the anaerobic–aerobic cycle. (**a**): LB-EPS; (**b**): TB-EPS.

**Figure 7 molecules-24-03358-f007:**
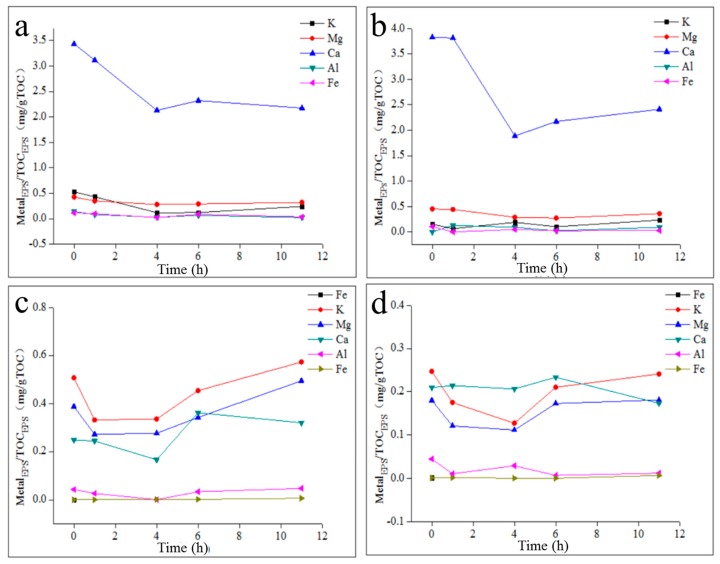
The content and variation of metal elements and TP in the TB-EPS during the anaerobic–aerobic cycle. (**a**): TB-EPS from a 35 °C sodium acetate reactor; (**b**): TB-EPS from a 35 °C sodium propionate reactor; (**c**): TB-EPS from a 20 °C sodium acetate reactor; (**d**): TB-EPS from a 20 °C sodium propionate reactor.

**Figure 8 molecules-24-03358-f008:**
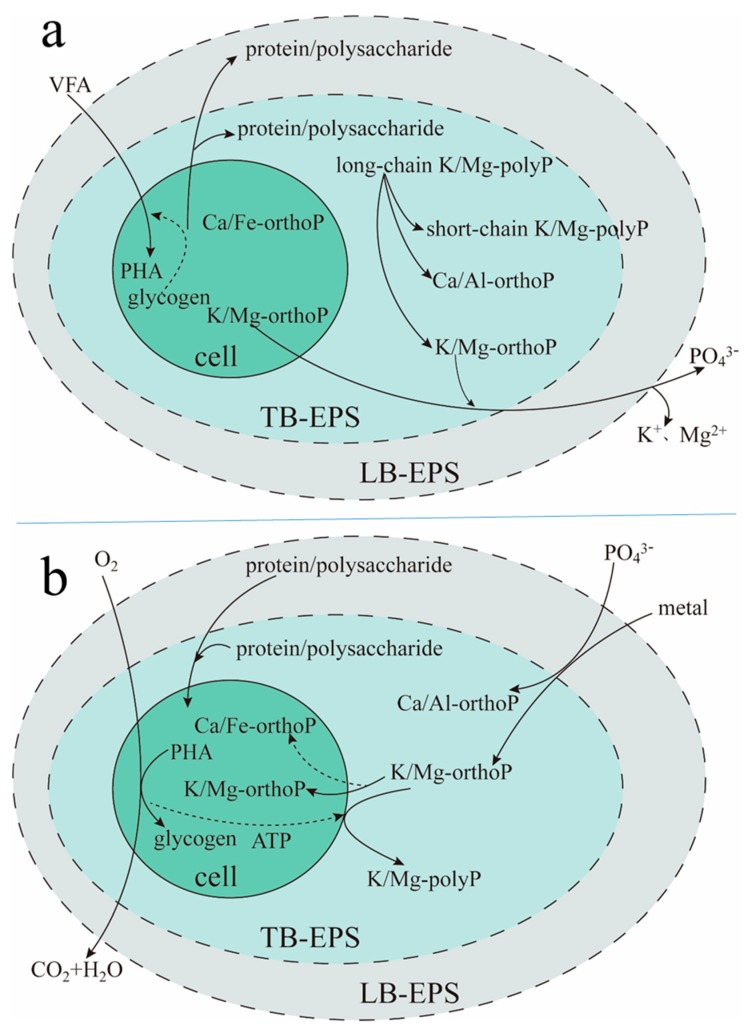
Supposed metabolic model for enhanced biological phosphorus removal (EBPR) with a consideration of the roles of TB-EPS and LB-EPS. (**a**): Anaerobic; (**b**): aerobic. Abbreviation: PHA: polyhydroxyalkanoates.

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
