# Peer review of "Layered Extraction and Adsorption Performance of Extracellular Polymeric Substances from Activated Sludge in the Enhanced Biological Phosphorus Removal Process"

_molecules, 2019, doi:10.3390/molecules24183358_

Round 1

Reviewer 1 Report

The authors provided answers for all the points of the reviewers, thus the ms has been significantly improved.

Just a short comment on the interpretation of the FTIR specra (fig. 5): on the x axis, the wavenumbers should appear in decreasing order, from 4000 to 400 cm-1.

Reviewer 2 Report

Authors have improve their article and also they have included more data.

Author Response

Thank you very much for your review.

This manuscript is a resubmission of an earlier submission. The following is a list of the peer review reports and author responses from that submission.

Round 1

Reviewer 1 Report

The study by Li and Xi on phosphorous in various types of extracellular polymeric substances is straightforward, clearly presented, and the conclusions seem justified.

This reviewer has only minor comments:

Line 128: Only microorganisms with a conspicuous morphology can be identified at the genus level. Are Brevibacteria really that conspicuous?

L129: filamentous bacteria not in italics

Overall, the manuscript would benefit from careful editing for proper use of the English language (e.g. by a native speaker or professional service).

Reviewer 2 Report

The authors discuss the effect of combining sonication and cation exchange resin to extract extracellular polymeric substances from EBPR sludge. In fig.2 by adding the two processes alone sonification+CER is about equal to the combine one sonification-CER. For example at 20oC sodium acetate Son=24 CER=120 while Son-CER=146; that is, the combine method gains 2mg out of 418 or 0.5%. Is this important for the given process? It is not discussed.

Reviewer 3 Report

The authors aimed at the determination of the absorption performance of two types of EPS at different temperatures and using different carbon-sources. The topic is interesting, however, in my opinion there are several issues to be addressed.

First of all, I didn’t see any numerical result, or measurements, based on which the supposed metabolic model could be designed.

The description of the experiments is not clear enough, and I don’t see the reason of the use of two different carbon sources (sodium acetate and sodium propionate), if their results are not compared. The authors investigated the effect of the temperature, but not that of the carbon source.

I suggest putting the two figures from the supplementary material to the main manuscript, as they are described in details in the text.

In section 2.3, where the FTIR results are described, there are no assignations for the mentioned phosphorus-containing and sulfur-containing (and not containing-phosphorus/sulfur) groups. And also, for the evidence of the carboxyl group in the IR spectra, a sharp peak around 1700 cm-1 should appear for the C=O bond.

Please reconsider the explanation in section 2.4. What would be good? If the different EPS complex a large amount of metal, or of not? Except the Ca at the 35°C reactors, all the other metals have almost the same concentration. Then how one can conclude that at 35°C the EBPR performance is poor, while at 20°C it’s OK?

A short description about the aerobic and ananerobic circumstances should be added.

Other comments:

Please do not start sentences with “And” The details of the extraction should be deleted from the caption of fig. 1 Please always write in full the abbreviations at their first appearance (e.g. VSS, TP, SEM, VFA, PHA, etc.) In section 3.2.2. line 256, I think the pore diameter is not correct, please check it

As a conclusion, I’m sorry to say, but I do not recommend this paper for publication.